# UV-Nanoimprint and Deep Reactive Ion Etching of High Efficiency Silicon Metalenses: High Throughput at Low Cost with Excellent Resolution and Repeatability

**DOI:** 10.3390/nano13030436

**Published:** 2023-01-20

**Authors:** Christopher A. Dirdal, Karolina Milenko, Anand Summanwar, Firehun T. Dullo, Paul C. V. Thrane, Oana Rasoga, Andrei M. Avram, Adrian Dinescu, Angela M. Baracu

**Affiliations:** 1SINTEF Microsystems and Nanotechnology, Gaustadalleen 23C, 0737 Oslo, Norway; 2National Institute of Materials Physics, Atomistilor Street 405 A, 077125 Magurele, Romania; 3National Institute for Research and Development in Microtechnologies-IMT Bucharest, Erou Iancu Nicolae Street 126A, 077190 Voluntari, Romania

**Keywords:** metasurfaces, photonic sensors, nanopatterning

## Abstract

As metasurfaces begin to find industrial applications there is a need to develop scalable and cost-effective fabrication techniques which offer sub-100 nm resolution while providing high throughput and large area patterning. Here we demonstrate the use of UV-Nanoimprint Lithography and Deep Reactive Ion Etching (Bosch and Cryogenic) towards this goal. Robust processes are described for the fabrication of silicon rectangular pillars of high pattern fidelity. To demonstrate the quality of the structures, metasurface lenses, which demonstrate diffraction limited focusing and close to theoretical efficiency for NIR wavelengths λ ∈ (1.3 μm, 1.6 μm), are fabricated. We demonstrate a process which removes the characteristic sidewall surface roughness of the Bosch process, allowing for smooth 90-degree vertical sidewalls. We also demonstrate that the optical performance of the metasurface lenses is not affected adversely in the case of Bosch sidewall surface roughness with 45 nm indentations (or scallops). Next steps of development are defined for achieving full wafer coverage.

## 1. Introduction

The past decade has seen myriad novel optical functions become realized through the exploration of enhanced light control arising from flat, ultrathin arrays of subwavelength nanostructures [1,2,3,4,5,6,7,8]. Such nano-photonic arrays are often called optical metasurfaces (OMS), owing to their display of optical properties beyond those typically found in nature. Examples of optical devices that have been demonstrated include innovative gratings, flat lenses, mirrors, advanced holograms, waveplates, polarizers, and spectral filters [3,9,10,11,12,13,14]. Following an exploratory phase of research typically conducted in university environments (resulting in more than ten thousand publications [15]), the technologies have started to migrate into industrial research with large multinational companies such as Samsung and ST Microelectronics running internal research [16] and even launching OMS products [17]. An important reason for this is that OMS can be multifunctional: integrating several functionalities into a single surface where conventional optical systems require several elements [18], thereby simplifying the complexity of the optical components. Furthermore, OMS are planar and can be integrated into existing production lines of e.g., image detectors and light sources such as LEDs [19] and VCSELs [20]. A challenge is, however, to achieve scalable fabrication that allows for sub-100 nm resolution patterning while ensuring high throughput and reproducible production [21].

Recently, Deep Ultraviolet (DUV) lithography has been suggested for commercial fabrication of OMS [22,23,24]. This technique is already widely used in fabs for micro-electronics such as microprocessors, and indeed DUV immersion stepper lithography is used to fabricate the first commercial metasurface lenses on 300 mm wafers by ST Microelectronics in collaboration with Metalenz [17]. However, DUV stepper technologies for sub 100 nm resolution involves technical complexity (to achieve DUV illumination and diffraction limited resolution) with costs that currently only can be sustained by very high production volumes.

Nowadays, only certain types of optical components are produced in the same volumes as for instance microprocessors (e.g., LEDs and photodetectors for certain wavelengths), whereas there are many optical sensor applications for which attaining such production volumes is unrealistic: e.g., the wide variety of optical sensors for niche applications currently using diffractive optical elements (DOEs). For the optical sensor industry to benefit from the many innovations developed in the OMS field, it is therefore necessary that scalable nanopatterning techniques are developed which are feasible also for low and medium volume production. In fact, even when high volume production is envisioned it is often an advantage that the development can be carried out using a technology that can be scaled up from low volumes, as this allows for addressing niche markets on the way towards mass-production commercialization. 

Ultraviolet Nano Imprint Lithography (UV-NIL) is a strong candidate for low, medium and high volume industrial metasurface fabrication as it can offer sub 100 nm resolution and high throughput at low technical complexity: The pattern transfer is performed by mechanically pressing a template into photoresist resin, rather than by selective illumination with sub 100 nm resolution such as in DUV lithography. Consequently, the technique is found in university, R&D and industrial labs fabricating e.g., diffractive optical elements (DOEs). Therefore, UV-NIL represents a versatile platform for both development and all volumes of production of metasurface-based sensor technologies. The UV-NIL technique also has additional advantages in terms allowing 3D structures (such as slanted structures relevant for AR applications), likely being more suitable for wafers with significant topography (relevant for integration into MEMS fabrication) and is likely less restricted in patternable die sizes than DUV Stepper [25,26,27,28]. 

Although Nano Imprint Lithography (NIL) is a technique that has recently been intensively used and studied, there are relatively few publications related to its use in the manufacturing of metasurfaces [29,30,31,32,33,34,35,36,37]. Furthermore, the label of metasurfaces fabricated by nanoimprint has is in fact been applied to quite a wide range of varied structures, ranging from resonance tuned [36], light emitting [32], polarization selective metasurfaces [33] to diffraction-limited lenses limited for NIR domain [29,30,38], high contrast NIR imaging [34] or for augmented reality [37] (the latter two papers have used NIL rather than UV-NIL). Generally, it is dielectric metasurface structures of relevance for flat lenses that are particularly challenging to fabricate, as they often rely on medium to high aspect ratio structures. In terms of attaining good lens efficiencies in UV-NIL metalenses, the amount of earlier published work is narrowed more. As far as we know, at the moment: (1) Brière et al. [32] reported, for a metalens made by GaN pillars with height (h) = 415 nm and 400 nm operating wavelength, a focusing efficiency of 16% in the case of using selective area sublimation (SAS) and 8% in the case of using Reactive Ion Etching technique (RIE); (2) Lee et al. [37] reported for metalenses made of poly-crystalline silicon pillars (length (l) = 220 nm, width (w) = 60 nm, h = 100 nm and period (*p*) = 400 nm) onto a quartz wafer substrate efficiencies of approximatively 79% for the co-polarized transmission at the wavelengths in the red, green and blue color domain, but lower efficiencies for the cross-polarized transmission reaching 12, 9 and 2.5% for the same wavelength regions compared to the calculated ones (29, 6 and 5%). (3) Yoon et al. [34] have reported a metalens, fabricated using a synthetized nanocomposite by dispersing Si nanoparticles a printable resin to create pillars with varying diameter from 260 to 650 nm, h = 1.2 µm and *p* = 900 nm operating at 940 nm wavelength, with 47% focusing efficiency; (4) Einck et al. [35] have fabricated metalenses working in the visible domain consisting in 600 nm tall hexagonal or rectangular nanoposts for TiO_2_-based nanoparticle ink, reporting an efficiency of approximatively 57% for the hexagonal ones and maximum 55% for the rectangular ones. Finally, the authors of the present work (Dirdal et al. [29,30,38]) have previously reported the fabrication of diffraction-limited dielectric metalenses (l = 350 nm, w = 230 nm, h = 1200 nm and *p* = 835 nm) working at the 1.310 and 1.550 µm wavelengths made by silicon etched rectangular pillars reaching up to 39% focusing efficiency.

Challenges towards UV-NIL metasurface fabrication currently lie in developing suitable processes that simultaneously ensure sufficient resolution and aspect ratios in the photoresist patterns for high-quality metasurface (e.g., metalenses) together with high pattern fidelity over large wafer areas and suitable deep reactive ion etching (DRIE) processes that allow for transferring the patterns to the wafer substrate. In this publication, we present optimized UV-NIL and DRIE (Bosch and Cryogenic) processes for the fabrication of dielectric metasurface lenses in silicon (Si) operating for NIR wavelengths. Compared to our earlier work [29,30,38] resist-opening processes have been improved to remove residual resist films between the photoresist structures without deteriorating the etch mask profiles. The process is shown to handle unevenness in the residual layer film thickness of the photoresist, which is important in order to have a well-functioning etch mask over large wafer areas. Moreover, further etch development has reduced tapering effects in cryogenic DRIE. We also show experimentally that using fewer etch cycles in the Bosch process (thereby creating large washboard-type sidewall surface roughness in the metastructures) does not deteriorate the optical performance of the OMS. We also present a novel Bosch process with many etch cycles which yields metasurface pillars without the characteristic sidewall-surface roughness, i.e., giving smooth sidewalls without tapering.

## 2. Design

The metasurface design relies on silicon (Si) rectangular pillars etched into a Si substrate, which implement a lens function to normally incident circular polarized light by applying phases according to the geometric phase principle [39,40,41,42]: i.e. the portion of incident circularly polarized light which becomes cross-polarized after interaction with the metasurface attains a relative phase given by the relative rotational angle α of the rectangular pillar. The structure is intended to work within the NIR range λ ∈ (1.3 µm,1.7 µm). The target design geometries are found from maximizing the amount of transmitted cross-polarization light in Rigorously Coupled Wave Analysis (RCWA) simulations (for details see earlier work [29]), and are found to be: h=1200 nm, p=835 nm, w=230 nm and length l=354 nm for the target wavelength λ=1.55 µm. In order to account for various anticipated processing effects (e.g., Bosch DRIE scallop sizes, tolerances, sidewall tapering in a mask, undercut, etc) three variations in lateral dimensions are chosen for populating the UV-NIL master (see Table 1):

Each of the chips has a square patterned area of 1.5 mm × 1.5 mm. The rotations of the pillars within these chips are chosen in accordance with the phase map of a lens with focal length 10 mm above the center of each chip for the target wavelength λ=1.55 µm (see Figure 1).

Due to the Si-substrate, there is a considerable reflection at the Si-air interface where the metasurface array is placed. By Fresnel equations we expect around 31% reflection at the Si-air interface, making the theoretical efficiency of the metasurface lenses around 69% (assuming a perfect AR coating on the backside of the substrate). The theoretical efficiency can be raised to above 90% by instead processing Si pillars on a quartz substrate, as has been carried out in [43]. Note however, that utilizing a quartz substrate changes the properties of the RIE etch and must be accounted for in the development of a fabrication process.

## 3. Metasurface Fabrication

### 3.1. UV-Nanolithography

The master wafer template was fabricated by using Electron Beam Lithography (EBL) and was purchased from NIL Technology (NILT, Haldo Topsøes Allé 1, 2800 Kongens Lyngby, Denmark). This template was then used for the fabrication of the stamp for the UV-NIL processing: GMN-PS90 silicone-based stamp material (purchased from OpTool, Rektorsgatan 3, 24762 Veberöd, Sweden) was spun onto the master wafer, and then placed into contact with a flexible carrier foil and cured using an EVG620 Smart NIL aligner. For the imprints, an adhesion layer mr-APS1 (Microresist GmbH, Berlin, Germany) was deposited on both (a) 6-inch (100) p type silicon wafer and (b) 4-inch (100) n type silicon-on-insulator (SOI) wafer with 30 µm Si device layer, followed by spin coating of mr-NIL210-200 nm resist (Microresist GmbH, Berlin, Germany), at 5000 rpm for 60 s. The different wafer diameters of (a) and (b) were chosen for compatibility with the two different etching equipment used to perform Bosch DRIE and Cryogenic DRIE, respectively. Afterwards, the nanoimprint process was carried out with an EVG620 mask aligner working at a constant time exposure mode. The exposure was performed with 31 mW/cm^2^ for 75 s.

A 2σ variation of around 40 nm in the resist thickness from wafer to wafer was observed, as well as around 2σ variation of 30 nm within a single wafer. We believe that this was caused by varying solvent vapor pressure in the spin bowl (solvents accumulate in the bowl when coating multiple wafers), and more controlled spin environments could reduce this issue. Nevertheless, the developed resist opening step (discussed next) effectively handled the varying thicknesses.

### 3.2. Deep Reactive Ion Etching

#### 3.2.1. The Bosch Process

The dry etch process we used consists of two steps. The first step is the anisotropic removal of the residual layer of the UV-NIL resist, and the second step is the actual Si etch. Both steps are performed in a Rapier process module by SPTS Technologies Ltd. (Newport, UK), which is optimized for running Bosch Si deep reactive ion etch (DRIE) processes. After imprinting and curing of UV-NIL resist, a residual layer (RL) of resist remains in the openings between the resist pillars, covering the substrate as shown in the cross-sectional SEM image Figure 2a. This RL needs to be removed (i.e., opening the RL layer between the resist pillars) prior to the Si DRIE process. In Figure 2a an RL thickness of around 110 nm is visible between the resist pillars. It should be noted that since the resist thickness varied from wafer to wafer prior to imprint, and at different positions on a single wafer, the RL thickness therefore also varied between fabricated samples. To handle this, the RL thickness was estimated for each imprinted wafer by measuring the resist thickness close to the target metalens chip using ellipsometry. The RL was removed by a continuous directional sputter-etch process using a low-pressure Ar plasma. The main process parameters for this step are in Table 2.

The RL removal step is run until the ellipsometer measurement shows that the RL has been completely removed. The typical time for this step is 150–300 s. Figure 2b shows the cross-section of the imprinted metalens after RL removal. The RL is fully removed while the metalens patterns in the UV-NIL resist are preserved with enough thickness for the Si DRIE process. By performing a sufficient overetch during the RL removal step we were consistently able to remove the RL within all chip variations and achieve good DRIE results, despite the initial variations in RL. This process stability is important for the aim of a full wafer population of metasurface lenses.

After RL removal, a Bosch DRIE process consisting of repeating cycles of passivation polymer deposition, de-passivation, and silicon etching was used to etch the pillar structures in silicon. Three such processes were developed with process parameters tuned to obtain three different degrees of characteristic washboard-like sidewall surface roughness, characterized by scallop sizes of 45 nm, 25 nm, and 10 nm (Figure 3). The main process parameters for these three processes are mentioned in Table 3.

The metalenses etched with Bosch DRIE process exhibit excellent vertical (90°) sidewalls without tapering (Figure 2). While scallops corresponding to the 25 nm and 45 nm process are clearly visible, the sidewalls of the pillars corresponding with the 10 nm process appear smooth. This somewhat surprising result is an interesting demonstration of the possibility of creating smooth-walled structures using the Bosch DRIE process. Chip A was used for the 45 nm process, Chip B was used for the 25 nm process and Chip C was used for the 10 nm process. 

#### 3.2.2. Cryogenic Deep Reactive Ion Etching

In our previous work, we reported the fabrication of a nanostructured metasurface comprising of rectangular nanopillars with variable rotations, fabricated both by e-beam lithography and UV-NIL [30,38] followed by cryogenic silicon etching. During the processing of the nano-pillar structures, the measured metalens efficiencies were found to be small compared to the simulations. Possible reasons for this include: (1) edge rounding of the rectangular profile, and (2) notch forming at the upper part of the nanopillars.

Regarding the first factor, the edge rounding of the rectangular profile of the nanopillars was due to the removal of the residual layer remaining after nanoimprint lithography. The removal of the residual layer was previously performed using O_2_ plasma, which resulted in an isotropic etch of the resist. The O_2_ radicals generated chemically react with the resist to form CO_2_, H_2_O, and other volatile chemical products, removing not only the thin residual layer, but also attacking the side-walls of the mask, thus resulting in a shrunken shape with rounded profiles. To mitigate this, the plasma etching recipe was modified to an Ar physical etch plasma. In this case, etching was performed entirely by Ar+ ions, which are accelerated by the DC bias applied on the lower electrode. Since the electrical field is perpendicular to the etched substrate, only horizontal surfaces are etched by physical sputtering, without significant damage to the patterned side-walls. The recipe used is presented in Table 4.

Regarding the second structural imperfection, the notch forming at the upper part of the nano-pillars can be caused by a low passivation regime during the first stages of the cryogenic etching process, a heat build-up due to inefficient heat transfer, or by a combination of both. To mitigate the low passivation regime, the cryogenic etch recipe was modified by introducing an O_2_/Ar plasma step in the first 5 s of the etching process to ensure the etching process starts in the passivation regime with no etching chemistry. Moreover, during the etch recipe, the ICP power was significantly reduced to reduce ion bombardment which in turn contributes to the heat build-up at the upper side of the nano-pillars. The balance between ICP power and RF power was adjusted to reduce the DC bias on the lower electrode to 35 V. A lower DC bias was found to result in inefficient removal of the oxyfluorosilicate passivation layer. The optimized cryogenic etching recipe is presented in Table 5 and the cross-sectional view of the nanopatterned silicon metalens after cryogenic DRIE process are shown in Figure 4.

After the metasurface lenses processing using Cryogenic Deep Reactive Ion Etching processes, Chip C obtained values similar to the target design geometries.

## 4. Optical Characterization

The optical characterization of the metasurface lenses was performed using a broadband source and a spectrometer in the 1100–1700 nm range. As shown in Figure 5, the metasurface lens focuses the collimated light (>Ø1.5 mm) into a fiber (Ø0.4 mm) that is connected to a spectrometer. The spectrometer records and provides the intensity of the collected light at different wavelengths. The cross-sectional area of the collimated beam is roughly 14 times larger than the cross-section of the fiber connected to the spectrometer. With an ideal lens, the entire cross-section of the collimated beam is focused onto the fiber, yielding 100% efficiency. Without any lens, however, only the cross-sectional area corresponding to the fiber is collected, i.e., giving an efficiency of around $\frac{0.4^{2}}{1.5^{2}}=7.1\%$. In other words, any efficiency measurement above this lower threshold indicates focusing. To analyze the efficiency of the metasurface lenses we have compared the intensity of the focused beam from the metasurface lenses to a standard aspheric lens (AL1210M-C, Thorlabs) which is close to ideal (i.e., close to 100% efficient). Unlike the standard aspheric lens, the metasurface lenses used in this work are designed for a particular polarization and do not have an anti-reflection coating. Thus, we have taken these two factors into account for our efficiency calculation.

Following the new processing steps presented in this paper, Figure 6 shows that the resulting metalenses etched with both Bosch and cryogenic DRIE processes give close to theoretical optical efficiency (50–52%). The Bosch metalens with smooth sidewalls and cryogenic lens are shifted relative to the simulation. This shift may be attributed to slightly smaller lateral dimensions of the Cryo and smooth walled Bosch pillars than targeted, as discussed in the next section. The simulation curve is based on the target dimensions outlined in Section 2.

Images of the diffraction limited focal points are shown in Figure 7a–c obtained by using a coherent laser source of 1.55 µm wavelength. Diffraction patterns characteristic of a square aperture are observed. The cross section of the normalized intensities is shown in Figure 7d: The distance from peak intensity to first extinction corresponds well with the diffraction limit (for wavelength λ=1.55 µm, focal length f=10 mm, lens dimension $D^{2}=1.5\;\mathrm{mm}\times 1.5\;\mathrm{mm}$) of $d=\frac{\lambda f}{D}\approx 10.3\;µm$, indicating that the metalenses provide diffraction limited focusing.

## 5. Discussion

The structures resulting from both the Bosch DRIE and Cryogenic DRIE processes have varying geometrical fidelity to the targeted perfectly rectangular pillars upon which the simulations were based (Figure 3 and Figure 4). The 10 nm scallop Bosch structures and the cryogenic structures appear rectangular with smooth sidewalls. The former exhibits 90° vertical sidewalls and sharp top edges, while the latter exhibits slight tapering and rounding off along the top edge. The 25 nm and 45 nm scallop Bosch structures differ from a rectangular pillar by their sidewall surface roughness (i.e., scallops). However, this difference does not lead to any significant consequences in terms of peak efficiency between the 45 nm scallop structures and the smooth-walled Bosch DRIE structures (Figure 6). As shown by simulations in our previous work [29] the surface roughness, in that it is strongly sub-wavelength, is not significant to the metasurface performance. There is however a horizontal shift between the displayed curves. This is in large part due to the differences in lateral dimensions between the structures. The scaling invariance of Maxwell’s equations dictates that decreasing all dimensions of a pillar equally is equivalent to blue shifting and downscaling the bandwidth of its efficiency curve. In the case of the fabricated metasurface structures, it is only the lateral dimensions that are decreased while the height is kept close to the targeted value of 1.2 µm. As a result, some differences in the shape of the curve occur. The same applies to the shift between the Cryo-structures relative to the Bosch-structures. Of the structures measured, it is the 45 nm scallop Bosch structures that have a bandwidth closest to the target structure. From the above reasoning we may infer that the effective dimensions of these pillars correspond well with the target structures, whereas the smooth-walled cryogenic and Bosch structures have lateral dimensions that are smaller than the target. The latter smooth walled structures can therefore be corrected by modifying the nominal dimensions in the master wafer template.

The high degree of structural fidelity observed in the SEM images between the fabricated smooth-walled structures and the targeted simulation structures, accounts for relatively high efficiencies achieved. The discrepancy in measured vs. simulated efficiency (∼53% measured vs. ∼64% simulated) may be accounted for by several possible loss mechanisms. One may expect diffraction losses and nearest neighbor coupling losses to arise from the metasurface structure not being a perfect array (contrary to what is assumed for the simulation curve). In addition, there may be measurement uncertainties such as differences in fiber-coupling between metasurface lenses and reference lens. Apart from such loss mechanisms, and as mentioned earlier (Section 2), the efficiency of the metalenses can be significantly increased if placed on a quartz substrate instead of Si. Then the transmission from the Fresnel equations across the boundary increases from 69% to 96%. This work has demonstrated metasurface structures of high structural fidelity relying on high throughput, large area patterning techniques: UV-Nanoimprint and DRIE. A challenge towards achieving full wafer coverage of metalenses with these techniques is the observed 2σ∼30 nm variation of resist thickness over the wafer. However, the resist opening processes employed here nonetheless succeeded in dealing with this variation for all the rectangular pillar metasurface structures which were in 3 chips placed at a radial range of r∈(0, 30 mm). As mentioned earlier, we believe more uniform resist film thickness can be achieved with automated dispense and perhaps control of the solvent vapor pressure. An advantage of the metasurface structures used in this study is that they consist of rotated but otherwise identical rectangles. This means that each unit cell has an equal filling factor, which simplifies the optimization of the UV-NIL process. That being said, this work demonstrated that chips containing rectangles of different sizes were all successfully fabricated (filling factors of 0.12, 0.17, and 0.24, corresponding to the chip dimensions of Table 1).

## 6. Conclusions

The presented UV-NIL and DRIE processes have succeeded in producing optical metasurface (OMS) lens structures that demonstrate good structural fidelity to the intended designs and optical efficiencies comparable to the simulated values. The processes developed are robust in handling observed resist thickness variations. Both Bosch and Cryogenic DRIE have been utilized. For both processes, the first step consists of opening the photo resist mask between the imprinted structures, i.e., the residual layer (RL). A physical Ar plasma etch was used to remove the RL on different tools (with differing parameters) for each of the DRIE methods.

Three different Bosch DRIE processes have been developed, targeting different amounts of the characteristic sidewall surface roughness (scallops) resulting from the cyclical and isotropic nature of the Bosch etch: 45 nm, 25 nm and 10 nm. The resulting structures exhibit excellent vertical (90°) sidewalls without tapering. Surprisingly the process aiming for 10 nm scallops results in sidewalls that appear to be smooth. This is an interesting demonstration of the possibility of creating smooth-walled surfaces while using the Bosch DRIE process. The Cryogenic DRIE etched structures also conform well to the target dimensions, with insignificant tapering present. All variants achieve high efficiencies in the range 50–52% compared to the theoretical upper limit of around 69% (Fresnel reflection at an intrinsic Si–air interface). Various loss mechanisms may account for the efficiency being less than ideal: Measurement uncertainty, structural imperfections, loss to higher order diffraction orders, absorption in Si. Replacing the Si substrate with e.g., quartz will increase the theoretical efficiency to above 90%.

## Figures and Tables

**Figure 1 nanomaterials-13-00436-f001:**
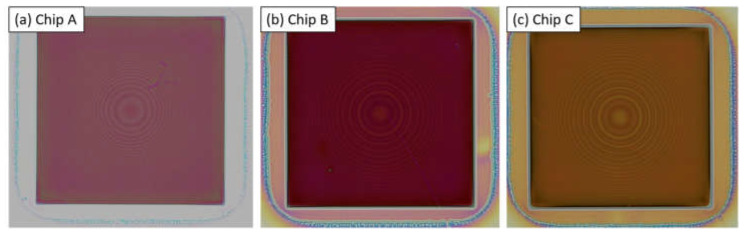
Microscope images of imprinted resist corresponding with (**a**) chip A, (**b**) chip B and (**c**) chip C. The centrosymmetric intensity variations are caused by varying diffraction due to different rotations of the resist pillars corresponding to the phase layout of a lens with focal distance 10 mm above the center of each chip.

**Figure 2 nanomaterials-13-00436-f002:**
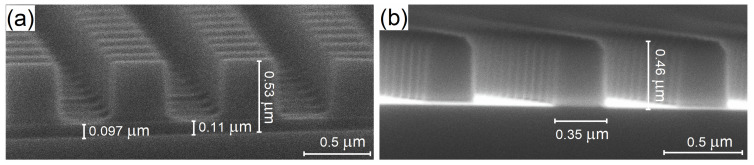
Cross-sectional SEM images of imprinted resist: (**a**) Chip A before the RL removal process, and (**b**) Chip C after the RL removal process.

**Figure 3 nanomaterials-13-00436-f003:**
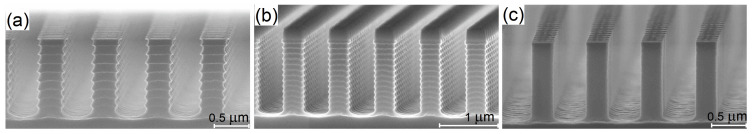
Cross-sectional SEM image of metalenses after Bosch DRIE process (**a**) 45 nm, (**b**) 25 nm, (**c**) 10 nm scallops.

**Figure 4 nanomaterials-13-00436-f004:**
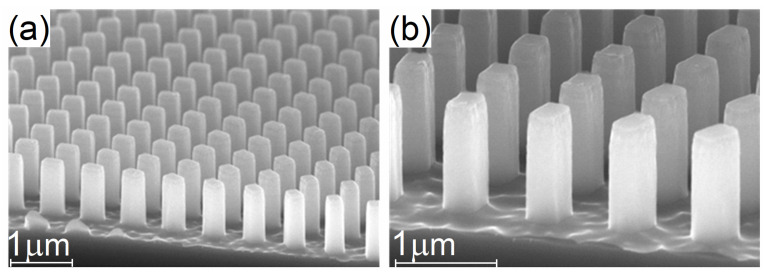
Cross-sectional SEM images of the nanopatterned silicon metalens after cryogenic DRIE process: (**a**) the fidelity of the rectangular nanopillars with (**b**) straight profile and smooth sidewalls.

**Figure 5 nanomaterials-13-00436-f005:**
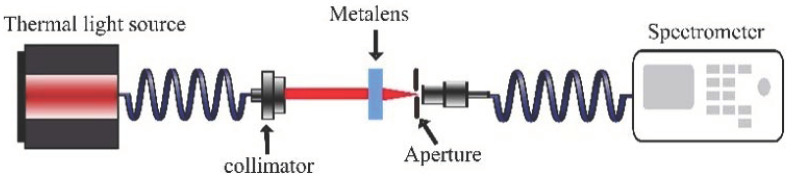
Schematic diagram of the optical setup for characterizing metasurface lens. The light from a thermal light source (SLS201/M, Thorlabs) is coupled into a multi-mode fiber (M76L01, Thorlabs) and collimated using a triplet collimator (TC06FC-1550, Thorlabs). The collimated light propagates toward the backside of the metasurface lens. The metasurface lens focuses the light beam into a multimode fiber (M28L01, Thorlabs) which is connected to a spectrometer (NIR-512, Ocean optics). The metasurface lens is attached to an X-Y-Z translation stage for alignment. The aperture (CPA1, Thorlabs) is placed before the fiber to block the stray light.

**Figure 6 nanomaterials-13-00436-f006:**
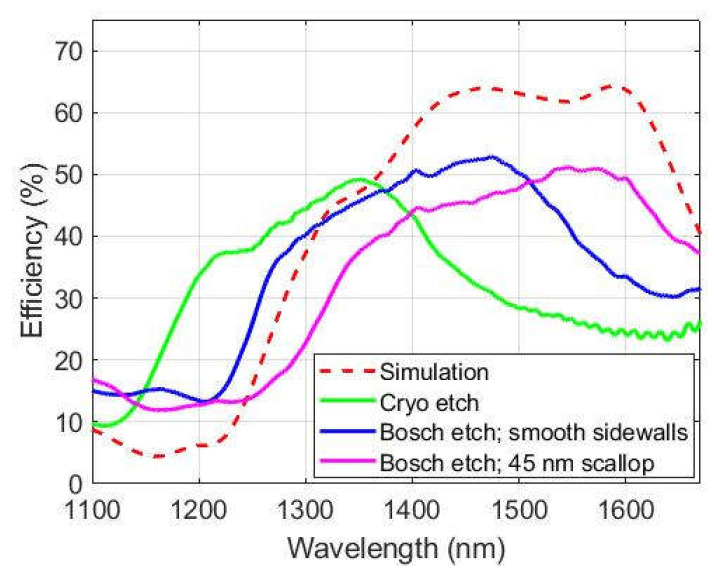
Simulation and measurement results of optical efficiencies for metasurface lenses. Simulation (dashed line) and measurements (solid lines) of the efficiency for both Bosch and cryogenic lenses are presented. The ripples in cryogenic lens measurements for larger wavelengths are due to the interference of light reflected from the oxide box of the SOI wafer. The simulation curve is made for perfectly rectangular pillars of the target dimensions.

**Figure 7 nanomaterials-13-00436-f007:**
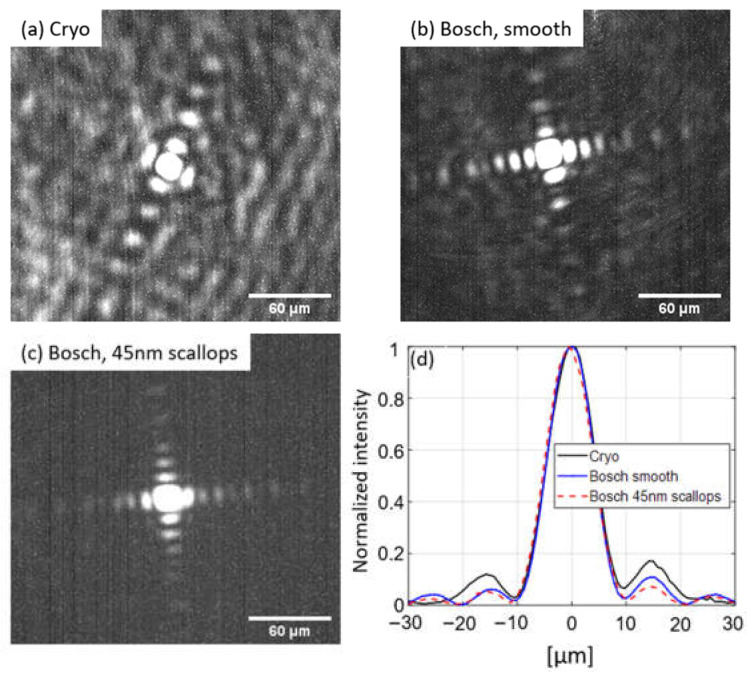
(**a**–**c**) Images of the focal points of the metasurface lenses. Diffraction patterns are observed corresponding to that expected of a square aperture (the metalenses are squares of 1.5 mm×1.5 mm). (**d**) Cross section of normalized intensity.

**Table 1 nanomaterials-13-00436-t001:** Lateral dimensions of the silicon pillars.

Chip	Pillar Width (nm)	Pillar Length (nm)
Chip A	351	475
Chip B	292	416
Chip C	237	361

**Table 2 nanomaterials-13-00436-t002:** Main process parameters for the RL removal step before Bosch DRIE.

Process Pressure (mTorr)	5
ICP Source Power (W)	1500
Platen Power (W)	30
Ar gas flow (sccm)	100
Substrate holder temperature (°C)	10
He backside pressure (Torr)	10
Process time (s)	Variable (150–300 s)

**Table 3 nanomaterials-13-00436-t003:** Bosch DRIE process parameters.

Scallop Sizes (nm)	45			25			10		
Total Etch Cycles	9			14			35		
Process Parameter	Step in Bosch Process			Step in Bosch Process			Step in Bosch Process		
	Passivation	Depassivation	Si etch	Passivation	Depassivation	Si etch	Passivation	Depassivation	Si etch
Process time (s)	1.5	1.0	1.8	1.5	1.0	1.0	1.0	1.0	1.0
Process Pressure (mTorr)	25	25	25	25	25	25	25	25	25
ICP Source Power (W)	2000	2000	2000	2000	2000	2000	1500	1500	1500
Platen Low Frequency RF Power (W)	0	130	30	0	130	30	0	50	25
Platen Power Duty Cycle (%)	20	20	20	20	20	20	20	20	20
C_4_F_8_ gas flow (sccm)	300	0	0	300	0	0	100	0	0
SF_6_ gas flow (sccm)	0	300	300	0	300	300	0	100	100
Substrate holder Temp. (°C)	10			10			10		

**Table 4 nanomaterials-13-00436-t004:** Main process parameters for the RL removal step before Cryogenic DRIE.

Process Pressure (mTorr)	7.5
RF Power (W)	200
Ar flow (sccm)	20
Substrate holder temperature (°C)	15
Process time (s)	60

**Table 5 nanomaterials-13-00436-t005:** Cryogenic DRIE process parameters.

Process Parameter	Step in Cryogenic Process	
	Ignition	Process
Process Pressure (mTorr)	7.5	7.5
ICP Source Power (W)	1000	1000
RF Power (W)	3	5
SF_6_ gas flow (sccm)	0	60
O_2_ flow (sccm)	10	8
Ar flow (sccm)	10	0
Substrate holder temp. (°C)	0	−115

## Data Availability

All data that support the findings of the study are provided in the main text. Raw data are available from the corresponding authors upon reasonable request.

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
