# Peer review of "UV-Nanoimprint and Deep Reactive Ion Etching of High Efficiency Silicon Metalenses: High Throughput at Low Cost with Excellent Resolution and Repeatability"

_nanomaterials, 2023, doi:10.3390/nano13030436_

Round 1

Reviewer 1 Report

In this paper, the authors describe a technology for creating a metalens in silicon with an efficiency of 50%, although the theoretical efficiency should be 69% for the near-IR range (1.3-1.6 µm). The technology includes UV-nanoimprinting and deep reactive ion etching and provides 100nm transverse resolution. This work continues the work of the authors [29, 30, 38], in which they already obtained an efficiency of 39%. The period of the metalens nanostructure was p=835 nm, and the relief height was h=1200 nm. The work can be published after the authors take into account the comments.

Comments

1. The numerical aperture of the metalens, its focal length and diameter are not indicated, and pictures (calculated and experimental) of the work of this metalens are not given. The work of the metalens is characterized only by Fig. 5, the dependence of the efficiency on the wavelength. But this does not prove that the lens was made and works like lenses, that is, it focuses light. In the experiment in Fig. 4, to determine the effectiveness, you can use not a metalens, but any other structure shown in Figures 1-3. The authors should add drawings confirming the operation of the metalens.

2. Authors should clarify what they mean by efficiency. Rather, they consider the amount of light that has passed through the structure to be effective. But it is correct to consider as efficiency not all the light that passed through the metalens, but only that which was focused into the focal spot. And this will be a significantly smaller value. The authors must indicate what part of the energy incident on the lens is concentrated in the focal spot.

Author Response

We thank the reviewer for insightful comments. We have responded in the attached document.

Reviewer 2 Report

It is a very nice paper about optimization of existing technology for achievement of higher quality results. The paper is very well written with very few mistakes and a clear story.

However, there are few discrepancies and I have some comments which may increase the message of the later published paper.

In table 1 you introduce three types "chips" with various dimensions of pillars.  Through the paper these "chips" are mentioned only roughly in the resist opening parts and at the end by claiming that all three fill factors were successfully fabricated.  I would expect at least in figure captions a notice about the type of chip it is regarded to. Especially for the optical measurements and simulation is this information essential.

There is a discrepancy between text in content and caption, and what I see at the Figure. 5. The text says: For Bosch metalenses, there is a shift in wavelength compared to the cryogenic lens and simulation. But obviously there is no shift between simulation and bosch 45 scallop, and a huge shift between simulation and cryo etch. This has to be better explained or corrected.

In the text and and conclusion you suggest that replacing Si substrate with quartz will lead to theoretical efficiency above 90%. As this is a manuscript dealing with fabrication technology, it would be wise to introduce problems of changing the substrate. I can imagine, that for quartz the dimensions of the pillars will change (increase), the DRIE parameters will change due to etching speed, maybe the working gas and the most importantly, the quality of the pillars after etching. Did you try this process on quartz substrates where you can predict the quality of the results? If yes, it should be included in the message, because pointing out only fact, that quartz has lower refractive index is misguiding and such information should not be presented in the paper. I even does not need to be in the paper, the work on Si substrate is sufficient.

After revising of my comments I will suggest to accept the manuscript for publication.

Author Response

We would like to thank the reviewer for insightful comments that have helped to improve the manuscript. We have added our response in the attached letter. 
